

# Where have I got to? Associations of age at marriage with marital household assets in educated and uneducated women in lowland Nepal

Akanksha A. Marphatia[1], Naomi M. Saville[2], Dharma S. Manandhar[3], Mario Cortina-Borja[1] and Jonathan C. K. Wells[1]

[1] Population, Policy and Practice Department, University College London Great Ormond Street Institute of Child Health, London, United Kingdom
[2] Institute for Global Health, University College London, London, United Kingdom
[3] Mother and Infant Research Activities, Kathmandu, Nepal

Corresponding author
Jonathan C. K. Wells,
jonathan.wells@ucl.ac.uk

## ABSTRACT

**Background:** Women's underage marriage (<18 years) is associated with adverse maternal and child health outcomes. Poverty in the natal household has been widely considered to be a key risk factor for underage marriage, but the evidence base is unreliable. When investigating this issue, most studies use marital wealth inappropriately, as a proxy for wealth in the natal household. In contrast, we investigated whether the timing of women's marriage was associated with the wealth of the households they marry into, and how this may vary by women's education level. This approach allows us to explore a different set of research questions which help to understand the economic value placed on the timing of women's marriage.
**Methods:** We used data on 3,102 women aged 12–34 years, surveyed within 1 year of marriage, from the cluster-randomized Low Birth Weight South Asia Trial in lowland rural Nepal. Linear mixed-effects regression models investigated independent associations of women's marriage age and education level with marital household wealth, and their interactive effects. Models adjusted for marital household traits. We analysed the full sample, and then only the uneducated women, who comprised a substantial proportion in our sample.
**Results:** In the full sample, we found that each later year of women's marriage was associated with 1.5% lower asset score for those with primary education, and with 0.3% and 1.3% higher asset score for those with lower secondary or secondary/higher education, respectively. For uneducated women, relative to marrying ≤14 years, marrying at 15, 16, 17 and ≥18 years was associated with 1.5%, 4.4%, 2.4% and 6.2% greater marital asset score respectively.
**Conclusion:** On average, marrying ≥18 years was associated with greater marital assets for secondary-educated women. There were only very modest benefits in terms of marital household wealth for delaying marriage beyond 16 years for uneducated women or those with low education. These findings elucidate potential trade-offs faced by families, including decisions over how much education, if any, to provide to daughters. They may help to understand the economic rationale underpinning the timing of marriage, and why early marriage remains common despite efforts to delay it.

## INTRODUCTION

For women, marrying before the age of 18 years, or underage, is associated with a range of adverse outcomes. These include less education, undernutrition, lower access to contraception, early childbearing, and higher morbidity and mortality during pregnancy and labour (*Ganchimeg et al., 2014*; *Godha, Hotchkiss & Gage, 2013*; *Goli, Rammohan & Singh, 2015*; *Marphatia et al., 2021b*; *Marphatia, Amable & Reid, 2017*; *Wells et al., 2022a*, *2021*). Many of these disadvantages are propagated to the next generation, where the cycle may repeat (*Bates, Maselko & Schuler, 2007*; *Fall et al., 2016*; *Finlay, Özaltin & Canning, 2011*; *Miller et al., 2022*; *Wells et al., 2022a*, *2022b*).

Despite growing recognition of these adverse consequences, 19% of women aged 20–24 years worldwide were married <18 years (*UNICEF, 2023*). This high prevalence is of major concern, indicating widespread violation of the rights of children. Early marriage hinders girls and adolescents from attaining general, sexual and reproductive health, and from safely transitioning to womanhood (*Lloyd, 2005*; *UN General Assembly, 1948*, *1979*, *1989*, *2014*). Underage marriage also reflects patriarchal power structures in many societies that restrict women to reproductive spheres and minimize their control over their own bodies (*Donath, 2000*; *Power, 2004*). The poor efficacy of interventions mean that achievement of the 5[th] Sustainable Development Goal, ending child marriage by 2030, is highly unlikely (*UN General Assembly, 2015*; *UNICEF, 2022*). Paradoxically, the prevention of early marriage has received little consideration in feminist economics studies, though it is attracting more attention in public health research (*Jayawardana, 2022*; *John, 2021*; *Roychowdhury & Dhamija, 2021*; *Wodon, Nguyen & Tsimpo, 2016*).

From an economic perspective, poverty in the natal household is widely considered to be a key risk factor for girls' underage marriage (*Otoo-Oyortey & Pobi, 2003*; *Pandey, 2017*; *Paul, 2019*; *Samuels et al., 2017*), and this understanding has guided interventions (*Amin et al., 2016*; *Nanda et al., 2015*). However, this association is not straight-forward, in part because of a poorly recognized limitation characterising most research studies. To test whether poverty predisposes to underage marriage, most studies assess household assets as the marker of wealth using Demographic Household Survey (DHS) data on ever-married women (*Rutstein & Johnson, 2004*). Clearly, women will only be assigned a marriage age if they are already married. Thus, wealth is actually measured in the marital household, and is then widely used as a proxy for wealth in the natal household (*Delprato et al., 2015*; *Male & Wodon, 2016*; *Sekine & Hodgkin, 2017*). When acknowledged explicitly, the rationale for this approach is that women marry into households with asset levels similar to those of the natal household (*Adams & Andrew, 2019*; *Male & Wodon, 2016*). This assumption remains untested, as no survey has measured wealth in both natal and marital households.

Evaluating marital household assets to test the hypothesis that poverty of the natal household predicts women's underage marriage is highly problematic. First, it is inappropriate to use an exposure (marital wealth) that occurs after the outcome (marriage)

to predict the timing of that outcome. In this case, the assessment of wealth reflects where women have ended up *after* marriage, and not where they came from *before* they married. Second, household assets may change after marriage, especially if there is a long time-gap between marriage and when the assets were quantified. Given these issues, the true relationship between wealth and women's marriage age remains poorly understood (*Schaffnit, Urassa & Lawson, 2019*), and the literature merits careful scrutiny.

## Literature review

Findings from previous studies in low-and middle-income countries that have inappropriately used marital household wealth as a proxy for natal household wealth have shown inconsistent findings. In Nepal, for example, *Pandey (2017)* used marital family wealth and other socio-demographic characteristics (*e.g.*, women's and husband's education, caste affiliation, geographical region) to investigate the odds of women marrying before/after 16 and 20 years of age. The results demonstrated no association of wealth with early marriage. In contrast, across South Asia, *Scott et al. (2021)* found that women from poor rural households were more likely to be married and have children at a young age. However, since wealth was measured in the marital household, these data indicate that poorer households are most likely to recruit in an underage wife, and do not demonstrate that poverty in the natal household drives families to marry their daughters early.

The few studies which correctly measured wealth in the natal household and quantified its association with age at marriage also do not consistently associate poverty with early marriage (*Adams & Andrew, 2019*; *Bajracharya & Amin, 2012*; *Marphatia et al., 2021c*, *2022b*; *Singh & Revollo, 2016*). In our own study population, for example, women's lower educational attainment rather than natal household poverty was associated with marrying early (*Marphatia et al., 2021c*).

Other studies that used marital wealth data appropriately have compared the proportion of women married underage by marital household wealth quintiles (*Guragain et al., 2017*; *MacQuarrie, 2016*; *Male & Wodon, 2016*; *Mathur, Greene & Malhotra, 2003*; *Mensch, Singh & Casterline, 2005*; *Paul, 2019*; *UNFPA, 2012*; *UNICEF, 2005*, *2021*). Analysis of DHS and Multiple Indicator Cluster Survey data (2000–2010) from 78 low-and middle-income countries on women aged 20–24 years found that the poorest 20% of households generally had the largest proportion of wives married underage (*UNFPA, 2012*). However, as noted by *Mensch, Singh & Casterline, 2005*, these studies can identify differences in early marriage rates in groups, and monitor changes in prevalence, but do not address why these patterns manifest at the individual level.

In order to understand marriage patterns in more detail, we need to consider the factors that shape the underlying decisions. In the cultural context of the Maithili-speaking Madhesi population of our study, where women have very low levels of autonomy, arranged marriages continue to be the norm (*Gram et al., 2018*; *Morrison et al., 2018*). In Nepal in general, some studies report that 'love' marriages are increasing. For example, data from 2012 onwards indicate that 32% of people married for love nationwide, and 50% in Chitwan (*Allendorf & Thornton, 2015*; *Human Rights Watch, 2016*). However, love

marriages remain rare in our study population, and instead, evidence from grey literature suggests that the timing of marriage decisions continues to be shaped by the preferences and characteristics of both natal and prospective marital households (*Human Rights Watch, 2016*).

In many South Asian societies, marriage decisions relate to several factors (*Marphatia et al., 2020a*, *2022a*; *Raj et al., 2014*). These include (a) the perceived costs to the natal household of maintaining a daughter, such as providing food, clothes and education; (b) the perceived benefits to the natal household, which include her continuing to undertake unpaid care work; and (c) the perceived benefits to the marital household of gaining a daughter-in-law at a particular age and education level (*Human Rights Watch, 2016*; *Jeffrey & Jeffery, 1994*; *Maharjan et al., 2012*; *Mathur, Malhotra & Mehta, 2001*; *Samuels et al., 2017*; *Verma, Sinha & Khanna, 2013*). Education plays a particularly complex role. A household's decision to invest in their daughter's education depends on their desire or economic capacity to send her to school, as even public schools incur material costs, *e.g.*, learning materials, and her reduced contribution to household subsistence. However, women's education may also be leveraged in marital matches to marry more educated boys (*Fafchamps & Shilpi, 2011*) and potentially also into wealthier households (*Jackson, 2012*). In South Asia, though illegal, it is common for the girl's family to pay a dowry to the husband, and education plays an important role in the bargaining process as the amount payable tends to increase in association with the girl's age and schooling (*Anderson, 2007*; *Field & Ambrus, 2008*; *Jeffrey & Jeffery, 1994*; *Mensch, Singh & Casterline, 2005*; *Sah, 2012*).

In summary, the investigation of how wealth is associated with women's marriage age is fraught with definitional and data-related issues. In many settings, the girl herself has limited agency over the timing of her marriage. Instead, marriage decisions are largely the consequence of others expressing their own interests. Misinterpreting the source of these interests hinders us from developing a comprehensive perspective on the practice of underage marriage which bridges feminist economics and public health. A feminist economics perspective provides a critical analysis of patriarchal social norms linking women's low levels of education, unpaid care work and the household economy, all of which shape underage marriage (*John, 2021*; *Urban & Pürckhauer, 2016*). Applying this perspective to public health interventions may help to explain why conditional cash transfers have not substantially delayed marriage (*John, 2021*; *Nanda et al., 2015*).

Understanding the association of age at marriage with wealth in the marital household is nevertheless an important question, because natal families are very aware of the wealth of a prospective marital household at the time the marriage is being arranged. We have previously used the same study design to investigate how women's marriage age is associated with the education of the husband (*Marphatia et al., 2021a*). Again, like wealth, the husband's education is readily observable to the natal household during marriage negotiations.

## Conceptual framework and hypotheses

We cite this previous literature to emphasise that our own study uses marital household assets more appropriately, as a marker of wealth in the household it was actually measured.

**(A) Previous research**

| Exposure | | Outcome |
|---|---|---|
| Marital household wealth used as a proxy for natal household's wealth | risk factor → | Girl's early marriage |

**(B) This study**

| Exposure | | Outcome |
|---|---|---|
| Variability in girl's age at marriage | associated with → | Marital household wealth measured in marital home |

varies by ↑

Girl's education level

**Figure 1 Conceptual framework.** (A) Illustrates the approach used by previous research. Most studies use the score of assets measured in the marital home inappropriately as a proxy for the natal household's wealth to investigate whether natal poverty increases the risk of early marriage. (B) Illustrates the approach used by our study. We use the score of assets more robustly, in the household (marital) where they were measured. We investigate whether variability in girl's age at marriage is associated with the score of assets in the marital home and how this association may vary by girl's education level.

Our approach allows us to explore a different set of research questions which, compared to previous studies, are more appropriate for data on marital household wealth.

Figure 1 sets our approach in context. Panel A shows that most previous studies have investigated whether poverty in the natal household (inappropriately measured in women's marital home) is a risk factor for early marriage. Panel B illustrates the goals of our study, which are to investigate whether women's characteristics at point of marriage (age and education) are associated with the wealth of the households they marry into.

This distinction, between previous studies and our approach, is important and enables us to emphasize that due to methodological flaws, these different questions have in fact been widely confused as detailed below. Conceptually, our study does not investigate where women came from in terms of their circumstances, but rather where they have arrived at marriage. We therefore cannot know if women entered into better circumstances at marriage.

Our study addresses a new hypothesis, which no previous study has systematically investigated:

Hypothesis 1: later marriage age is associated with an increased marital household asset score;

However, as discussed above, Hypothesis 1 requires probing in greater detail, because the association of marriage with marital wealth may not be uniform, but may rather depend on the woman's level of education. We therefore also test the following hypotheses:

Hypothesis 2: higher educational attainment among women is associated with an increased marital household asset score;

Hypothesis 3: the association of marriage age with the marital household asset score varies according to the level of women's education;

We first test these hypotheses in the whole sample, and then explore the association of marriage age with assets specifically in uneducated women only, who comprise a substantial proportion of our population. This adds value to our approach because for uneducated women, the association of other predictor variables with marital household asset score is entirely independent of, or unconfounded/mediated by, their education.

If marriage decisions simultaneously consider the age and education of women on the one hand, and the level of assets in the marital household on the other, we might expect that women are married at an age which advantages them economically. We therefore test an additional hypothesis:

Hypothesis 4: for any given level of education, women are most commonly married at an age which would maximize marital household assets.

## MATERIALS AND METHODS

The data for this analysis come from the cluster randomized controlled (non-blinded) Low Birth Weight South Asia Trial (LBWSAT). This trial was conducted across 80 Village Development Committees (VDCs) in the southern areas of Dhanusha and Mahottari districts, in Province 2 of the lowland Terai ecological zone of Nepal (*Saville et al., 2016*). Each VDC comprised one to several villages which were served by the same health facility. Of the 64,000 eligible married women who consented to menstrual monitoring between December 2013 and February 2015, a total of 24,682 pregnant women were recruited into LBWSAT (*Saville et al., 2016*). The trial assessed the impact of four different community interventions during pregnancy on birth weight and infant growth (weight-for-age z-score, from 0–16 months) (*Saville et al., 2016*).

The Nepal Health Research Council (108/2012) and University College London (UCL) Research Ethics Committee (4198/001) granted ethical approval to conduct the trial. VDC secretaries provided consent for inclusion of villages from lowland Nepal in the trial. Women gave written consent to participate in the trial. Guardians consented to the participation of married adolescents younger than the majority age of 18 years. Additional ethical approval was obtained from the Research Ethics Committee at UCL (0326/015), the University of Cambridge (1016), and the Nepal Health Research Council (292/2018) for secondary analyses of LBWSAT data.

### Variables

Our dependent (outcome) variable was marital household assets, expressed as a continuous score. Each household was assigned an 'asset score' based on the ownership of a range and number of consumer goods, and various components of household infrastructure. This standard approach to assets assessment is widely used by nationally representative surveys (*Rutstein & Johnson, 2004*; *Vyas & Kumaranayake, 2006*) and has been validated to be appropriate in our study context (*MOHP, New ERA & ICF International, 2017*). Assets were measured in the household where women were living when they enrolled in the trial, and were therefore measured in either the natal or marital

household, but not both. We selected only those households where the woman was living in their marital home, who comprised the majority (78%) of the sample. Since our asset score was measured in the marital household after, and not at, marriage, we controlled for the time since marriage by restricting the sample to women who had married in the past 12 months.

The marital household asset score used in our study was constructed from nine variables using principal component analysis (PCA) (*Filmer & Pritchett, 2001*; *Rutstein & Johnson, 2004*; *Vyas & Kumaranayake, 2006*). Compared to other methods for calculating wealth, the PCA is considered to be the most widely used, robust and comparable approach (*Poirier, Grépin & Grignon, 2020*; *Vyas & Kumaranayake, 2006*). These household assets represent relatively stable markers of wealth, relating to durable goods such as the structure of the home or the presence of goods or services that require significant financial outlay. Regarding the marital household, these assets were assumed already to exist before a girl marries into it. We excluded goods such as ownership of a colour television, motorbike, or computer, because they could have been acquired after marriage. The first principal component had positive factor loadings for all nine variables and accounted for 34.5% of the variability, compared to 12.7% from the second principal component. Thus, the first score was taken as the marker of wealth. A review article found that most studies using the PCA wealth index explained 12% to 27% of the variance (*Vyas & Kumaranayake, 2006*). In comparison, our PCA component 1 explained a higher proportion (34.5%) of the variance, making it a good proxy for household wealth.

The nine variables contributing the highest factor loadings to the first principal component, listed in order of decreasing size (weight shown in parenthesis), were: wall (0.767) and roofing materials (0.747), toilet facilities (0.701), flooring materials (0.700), number of rooms used for sleeping in the house (0.561), land ownership (0.484), access to electricity (0.430), drinking water source (0.399), and use of non-biomass cooking fuel (0.299). These weights reflect the contribution of each variable to the marker of wealth, and imply that the quality of the household's fabric is the most important aspect in the wealth index. All nine variables had strong positive correlations ($p < 0.001$) with husband's levels of education, ranging from 0.184 for roofing materials to 0.402 for toilet facilities. In our study population, assets were mostly male generated because males directly earn and control income. Due to seclusion norms, women are largely confined to the home; being extremely limited in their autonomy, young married women generally do not work for, or control, income (*Gram et al., 2017*, *2018*; *Maharjan & Sah, 2012*; *Morrison et al., 2018*). However, women's unpaid care work (*e.g.*, childcare, household work such as cleaning) nevertheless makes an invaluable contribution to sustaining the household and the well-being of all its members (*Marphatia & Moussié, 2013*).

Our explanatory (predictor) variables were women's marriage age and their educational attainment. Our study focused on a highly marginalised group of women, most of whom are illiterate, and therefore cannot precisely tell us their age, nor exactly when they were married. To address this, fieldworkers were trained on determining women's age as precisely as was possible, by asking numerous questions to verify their response. Women's marriage age was recorded as an integer value in running years because this was how

people generally reported age in this region of Nepal. We converted these ages to completed years (running years minus 1) for analysis. Marriage age was coded into five groups based on the distribution in our data: ≤14 years, 15 years, 16 years, 17 years and ≥18 years. We created these groups because we wanted to know whether later ages at marriage were associated with more assets in the marital home. The youngest and oldest marriage age groups were combined across years due to small numbers. Educational attainment (highest class completed in school) was coded into four levels based on the structure of the education system in Nepal (*Ministry of Education Nepal, 2016*), and also based on the distribution of data in our sample: none, primary (1–5 years), lower-secondary (6–8 years) and secondary or higher (≥9 years).

Since assets may also be shaped by other markers of social status, our models included husband's education (coded similarly to women's education) and caste. In Madhesi society, caste reflects diverse normative behaviours and values, as well as social constraints (*Bennett, Dahal & Govindasamy, 2008*; *Maharjan & Sah, 2012*). Caste was measured in the marital home, and, reflecting the distribution of the data, was described as: disadvantaged (Muslim or Dalit), middle (Janjati or other Terai groups), or advantaged (Brahmin or Yadav). We grouped Muslim and Dalit castes together because of their younger marriage age and lower education relative to the middle and advantaged castes. Ideally, we would have included additional variables relating to the wealth of the marital household, such as education of husband's parents and husband's (or head of household's) occupation and income, but our study did not collect these data.

## Statistical methods

We used median and interquartile range (IQR) to summarise continuous variables, given the skewed distributions of age-and timing-related data. We tested for biases in characteristics between women whose assets were measured in their natal household *versus* marital household. We used chi-squared tests for categorical variables. For continuous variables, we used the non-parametric *k* samples analysis of variance (Kruskal-Wallis test), to test for homogeneity of location. We used scatterplots to show crude associations of the raw marital household asset score against women's age at marriage, stratified by women's four educational attainment groups. Spearman's coefficient described the correlation between women's age at marriage and marital household asset score by women's different education groups. To improve interpretation of statistical models, we then standardized the asset score based on the first principal component of the durable assets data to a scale of 0 to 100. Coefficients thus showed the increase in the asset score corresponding to each category of marriage age, education, *etc*., where the intercept described the mean of women in the reference group for each predictor and the coefficients were deviations in percentage points from that mean, associated with other predictor values or categories.

To test our hypotheses, we fitted four linear mixed-effects regression models with a random effect on the intercept accounting for within-cluster variability. We assessed the models' assumptions regarding normality and homoscedasticity of the fixed-effects residuals, and of normality of the random effects by inspecting the corresponding plots. Then we tested this formally with the Shapiro-Wilk test, which showed a non-normal
distribution of our outcome variable. However, we still presented the linear mixed-effects models in the main article, because quantile regression showed similar results (Tables S3 and S4, Fig. S2). Quantile mixed-effects regression (Geraci, 2014) is an alternative approach. It works on the same scale as the outcome variable, and the interpretation of the coefficients is the same to linear mixed-effects models. The main difference between these two approaches is that the quantile regression estimates effects on the outcome variable's median (conditional on covariates), whereas linear regression estimates the effects on the mean.

In the full sample, linear mixed-effects models showed the separate associations of marital household asset score with women's marriage age (Model 1, hypothesis 1) and women's education level (Model 2, hypothesis 2). For hypothesis 3, we tested the combined interactive associations of women's marriage age and their education (Model 3), taking into account marital household traits (Model 4). The interactions between maternal age at marriage and their education were defined as the product of the maternal marriage age as a numerical value in five groups, similar to Model 1 (≤14 years, 15 years, 16 years, 17 years and ≥18 years) and the three education groups ('1–5 years', '6–8 years' and '≥9 years'), relative to uneducated women as the reference group.

Hypothesis 3 also tested the separate associations of marital household asset score with women's marriage age for only the uneducated women (Model 1), taking into account marital household traits (Model 2). For hypothesis 4, we showed heat maps of the absolute number and percentage of women marrying at different ages by their level of education. This enabled us to examine the most common age for women of a particular education group to marry.

We used linear mixed-effects regression rather than Ordinary Least Squared Regression in order to account for clustering with a random effects term for the 80 geographic (VDC) clusters (which were the unit of randomisation in the trial). Models controlled for women's age, which, when included together with their age at marriage effectively accounted for the time-gap between when they married and their current age (when they were recruited into the trial). We used women marrying ≤14 years, lack of education and disadvantaged caste as the reference groups. We reported the regression models' coefficients and their standard errors (s.e.), and statistical significance as: *$p < 0.05$, **$p < 0.01$, and ***$p < 0.001$. We also give the $p$-value if <0.10. We evaluated goodness-of-fit and report both the marginal and conditional Nakagawa-Schielzeth conditional $R^2$. The conditional $R^2$ measured the percentage of variance explained by both the model's fixed and random effects, whereas the marginal $R^2$ considered only the variance of the fixed effects (Nakagawa & Schielzeth, 2013). Whilst we reported these $R^2$ values for information, we were not trying to explain individual-level variability in household wealth, but rather the overall pattern of association by women's marriage age, to test whether on average, later women's age at marriage was associated with greater wealth in the marital household. Models were fitted using the R library lme4 (Bates et al., 2014), lqmm (Geraci, 2014) and SPSS 26 (IBM Corp., Armonk, NY).

Although marriage age, education and assets may vary across trial arms, we did not expect this to bias our results and therefore did not adjust for them in models. This is
because the trial recruited already married and currently pregnant women, which means that the interventions could not have influenced marriage age or education. Moreover, assets were measured before the trial was conducted, so the cash supplementation arm could not have changed their value.

## RESULTS

### Sample selection

Of a total of 24,682 women recruited into the LBWSAT, we first excluded 1,235 women with missing data on assets altogether. Second, of these 23,447 women, we excluded 19,463 women who had been married for ≥1 year. We selected only women within 1 year of marriage to ensure a small time-gap between marriage and when the assets were quantified. Of these 3,984 women, we then excluded 874 women whose assets were measured in their natal home and 8 women who had no information on the household where assets were measured. Our sample for analysis therefore consisted of 3,102 women aged 12–34 years. Table S1 shows differences between women where assets were measured in their natal *vs.* the marital home, but the magnitudes of effect are small, and are not expected to bias our results.

### Description of sample

Our analysis contributes to understanding marriage strategies of the Maithili-speaking Madhesi population in the Nepal Terai. This is important because even among the earlier-marrying and lower-educated women in the Terai, the Maithili-speaking Madhesi women have the highest odds of not being in school, and of marrying before 16 years of age (*Marphatia et al., 2020a*; *Ministry of Education Nepal, UNICEF & UNESCO, 2016*; *Pandey, 2017*). Women in our sample had a median age of 17 years (IQR 2) (Table 1) and the median age at marriage was 16 years (IQR 2). Overall, 20% of women were married ≥18 years, 10% married ≤14 years, and 69% between 15–17 years, while 39% were uneducated, 12% had completed primary, 19% lower-secondary, and 30% secondary or higher education. On average, husbands had completed more education than their wives: 34% of husbands were uneducated, 12% had completed primary school, 19% lower-secondary and 35% secondary or higher education. Within our study population, 31% of marital households were from disadvantaged castes, 44% from the middle, and 24% from relatively advantaged groups.

After standardizing on a scale of 0–100, the median marital household asset score was 39.6 (IQR 32.1). The score was lowest for women who had married ≤14 years (36.9, IQR 25.1), and increased with marriage age, being highest for those who married ≥18 years, (42.2 IQR 38.4). Scatterplots in Figs. 2A–2D show the raw marital asset score against women's marriage age, stratified by women's educational attainment, for women who were married between ages 12–24 years to emphasize the association (Fig. S1 shows the entire range of marriage age). There was no linear trend between marital assets and marital age in (a) uneducated women (Spearman's coefficient 0.01, $p = 0.631$), those with (b) primary ($-0.06$, $p = 0.272$) or (c) lower-secondary education (0.01, $p = 0.868$), but a positive trend among those with (d) secondary or higher education (0.14, $p < 0.001$).
**Table 1 Descriptive statistics for women aged 12–34 years, surveyed within ≤1 year of marriage in lowland Nepal (LBWSAT) (*n* = 3,102).**

| Background characteristics | *Median (IQR)* |
|---|---|
| Age (years) | 17 (2) |
| Age at marriage (years) | 16 (2) |
| **Independent (predictor) variables** | *Frequencies (%)* |
| Marriage age (years) | |
| ≤14 years | 321 (10) |
| 15 years | 598 (19) |
| 16 years | 897 (29) |
| 17 years | 650 (21) |
| ≥18 years | 636 (20) |
| Women's education level (years) | |
| None | 1,223 (39) |
| Primary (1–5 years) | 360 (12) |
| Lower-secondary (6–8 years) | 580 (19) |
| Secondary or higher (≥9 years) | 939 (30) |
| Husband's education level (years) | |
| None | 1,065 (34) |
| Primary (1–5 years) | 362 (12) |
| Lower-secondary (6–8 years) | 576 (19) |
| Secondary or higher (≥9 years) | 1,099 (35) |
| Caste affiliation | |
| Disadvantaged: Dalit, Muslim | 974 (31) |
| Middle: Janjati, Terai castes | 1,371 (44) |
| Advantaged: Yadav, Brahmin | 757 (24) |
| **Dependent (outcome) variable** | |
| Marital household asset score | 39.6 (32.1) |
| Marital household asset score by women's marriage age | Asset score |
| Marriage age (years) | *Median (IQR)* |
| ≤14 years | 36.9 (25.1) |
| 15 years | 37.2 (25.6) |
| 16 years | 40.5 (30.9) |
| 17 years | 40.5 (31.6) |
| ≥18 years | 42.2 (38.4) |

Note:
   *n, number*; IQR, Interquartile range; %, percentage.

## Hypotheses 1 to 3

Table 2 presents linear mixed-effects models of the full sample. Focusing only on women's marriage age (hypothesis 1), Model 1 shows that relative to those married ≤14 years, the asset score increased successively with later age at marriage, most noticeably from 16 years onwards. Compared to women married at ≤14 years, the asset score increased by 1.4% for those married at 15 years, 4.6% for those married at 16 years, 4.9% for those married at 17 years, and 9.1% for those married at ≥18 years. Overall, marriage age (together with

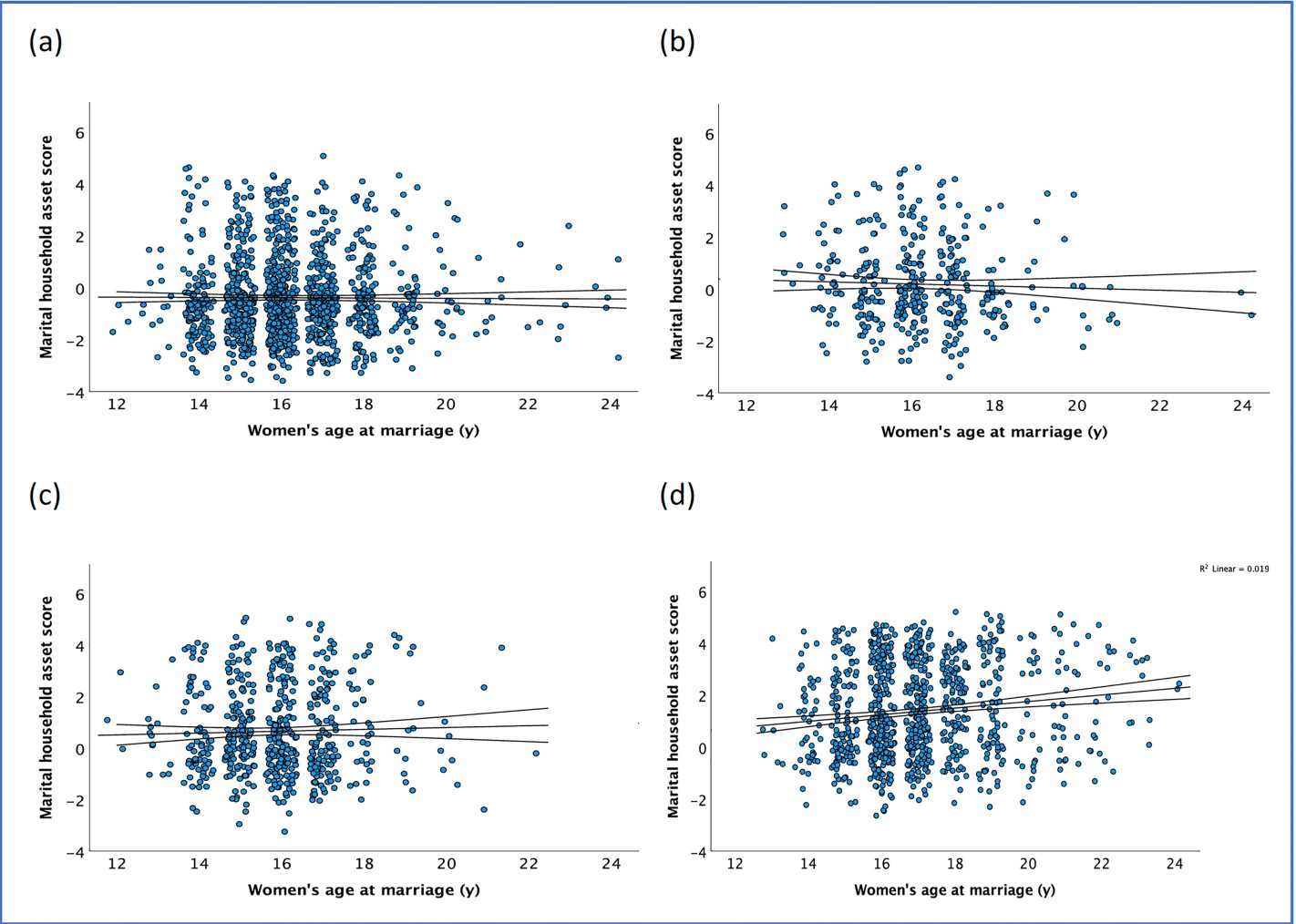

**Figure 2  Scatterplot of marital household asset score against women's marriage age, for women who married between ages 12–24 years, surveyed within ≤1 year of marriage, stratified by women's educational attainment ($n$ = 3,086)\*.** Plots use the raw values of the marital household asset score. (A) Uneducated women ($n$ = 1,211), (B) Primary education ($n$ = 359), (C) Lower-secondary education ($n$ = 580), and (D) Secondary education or higher ($n$ = 936). \*\*Marriage age is only available in completed integer years. Jitter has been added to show more data points. Lines represent confidence intervals of the regression slope. The scale for marriage age is set from 12 to 24 years to maintain consistency across the four plots. Figure S1 includes all women.                

women's current age) explained 10% of the variance in marital household asset score. Focusing only on women's education (hypothesis 2), Model 2 shows that in comparison to no education, primary, lower-secondary and higher education were associated with 6.8%, 12.7% and 20.3% greater asset score respectively, illustrating a strong dose-response association, as shown in Fig. 3. Overall, women's education (together with their current age) explained 23% of the variance in marital household asset score.

Model 3 (hypothesis 3) includes both education and marriage age plus interaction terms.

Independent of education, each higher age-group for marriage was associated with 0.6% greater asset score, while compared to no education, completing primary, lower secondary or secondary/higher education was associated with 11.3%, 12.1% and 13.0% greater asset

**Table 2 Linear mixed-effects models of women's marriage age and their education with marital household asset score for women aged 12 to 34 years, surveyed within ≤1 year of marriage (*n* = 3,102).**

| | Dependent variable = Marital household asset score | | | |
|---|---|---|---|---|
| | **Model 1 Women's marriage age** | **Model 2 Women's education** | **Model 3 Women's marriage age, their education, and interaction terms** | **Model 4 Women's marriage age, their education, marital household traits and interaction terms** |
| | *β (standard errors)* | *β (standard errors)* | *β (standard errors)* | *β (standard errors)* |
| Women's age (y) | −0.1 (0.4) | 0.7 (0.2)*** | 0.020 (0.3) | −0.002 (0.3) |
| Women's marriage age (y): ≤14 y | Reference | | | |
| 15 years | 1.4 (1.5) | | | |
| 16 years | 4.6 (1.6)** | | | |
| 17 years | 4.9 (1.8)** | | | |
| ≥18 years | 9.1 (2.5)*** | | | |
| Women's marriage age groups (y))[1] | | | 0.6 (0.6) | 0.7 (0.6) |
| Women's education (y): None | | Reference | Reference | Reference |
| Primary (1–5 years) | | 6.8 (1.2)*** | 11.3 (3.2)*** | 8.6 (3.1)** |
| Lower-secondary (6–8 years) | | 12.7 (1.0)*** | 12.1 (2.5)*** | 7.2 (2.5)** |
| Secondary or higher (≥9 years) | | 20.3 (0.9)*** | 13.0 (2.5)*** | 7.6 (2.4)** |
| Husband's education (y): None | | | | Reference |
| Primary (1–5 years) | | | | 5.1 (1.2)*** |
| Lower-secondary (6–8 years) | | | | 7.6 (1.0)*** |
| Secondary or higher (≥9 years) | | | | 13.5 (1.0)*** |
| Caste: disadvantaged | | | | Reference |
| Middle | | | | 0.9 (0.8) |
| Advantaged | | | | 1.7 (1.0))[2] |
| Interaction terms: uneducated women | | | Reference | Reference |
| Women's primary education and marriage age (y) | | | −1.5 (0.9) | −1.5 (0.9) |
| Women's lower-secondary education and marriage age (y) | | | 0.2 (0.8) | 0.3 (0.8) |
| Women's secondary education and marriage age (y) | | | 2.1 (0.7)** | 1.3 (0.7)* |
| Marginal *R*-squared | 0.02 | 0.17 | 0.17 | 0.22 |
| Conditional *R*-squared | 0.10 | 0.23 | 0.24 | 0.29 |

**Notes:**

*n*, number. Models include fixed and random effects estimates for geographic clusters.

* *p* < 0.05.
** *p* < 0.01.
*** *p* < 0.001.
[1] Coded similar to Model 1: ≤14 years, 15 years, 16 years, 17 years and ≥18 years.
[2] *p* = 0.096.

score respectively. However, there were additional interaction effects, whereby taking into account the above associations, each higher age-group for marriage was associated with 1.5% lower asset score for those with primary education, and with 0.2% and 2.1% higher

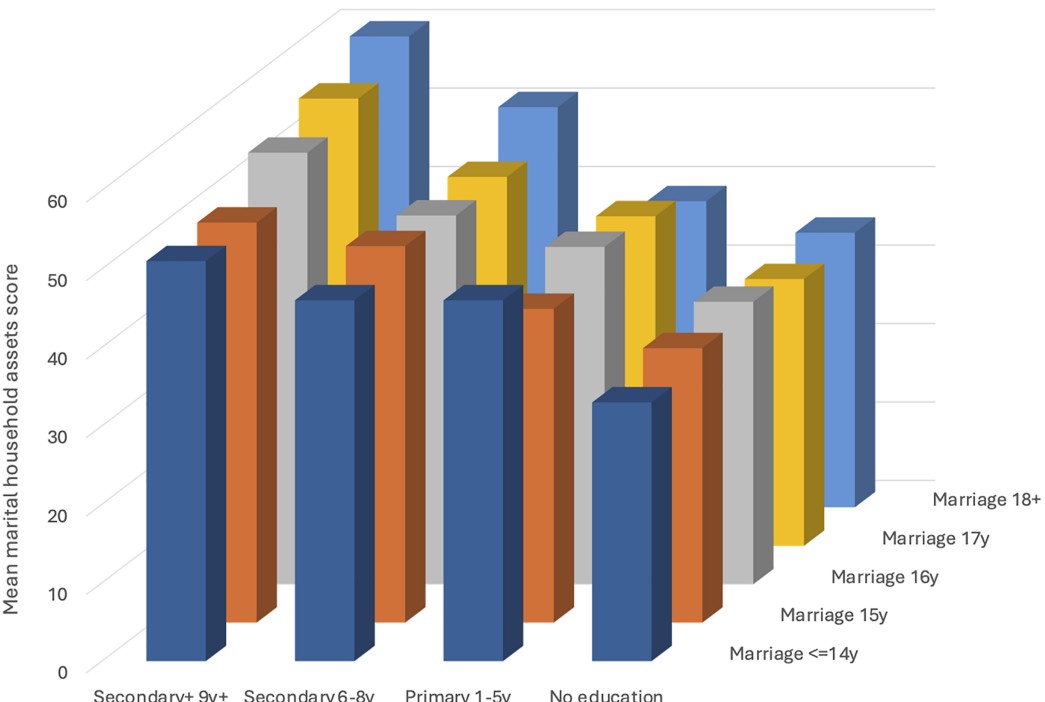

**Figure 3 Mean marital household assets score by women's education and age at marriage.** This 3-D plot illustrates the mean marital household assets score according to women's education level and age at marriage. For any given age at marriage, mean assets increase systematically with education level. However, for any given level of education, mean assets show negligible increases with age at marriage except for marriage at 18+ years for those with lower secondary education, and more systematically with later age at marriage for those with higher secondary education level.

asset score for those with lower secondary or secondary/higher education respectively. This model explained 24% of the variance in asset score.

Figure 4 shows the predicted values derived from Model 3 for marital household asset score by women's marriage age and education. The model assumes a linear change in assets by marriage age group, and includes random effects by cluster (Table S2). It shows that intercepts of all educated groups were higher than the intercept for the uneducated group. The slope for the uneducated group remained essentially flat, with very small changes across the marriage age groups. The slope for the primary-educated group (1–5 years) decreased until 15 years of marriage, and then remained constant. The slope for the lower-secondary educated group (6–8 years) increased after age at marriage of 17 years, while the slope for the secondary-educated group (≥9 years) showed a constant increase after 15 years. The increase in the asset score was almost 15% higher among women in the highest group of education compared to those who were uneducated for marrying at ≤14 years, and nearly 20% higher for marrying ≥18 years.

Table 2, Model 4 (hypothesis 3) shows the overall pattern remains after adjusting for husband's education and caste affiliation. The main difference in this model compared to Model 3 is a decrease in the coefficients for the three women's education groups, indicating that part of the association of women's education with household assets was mediated by

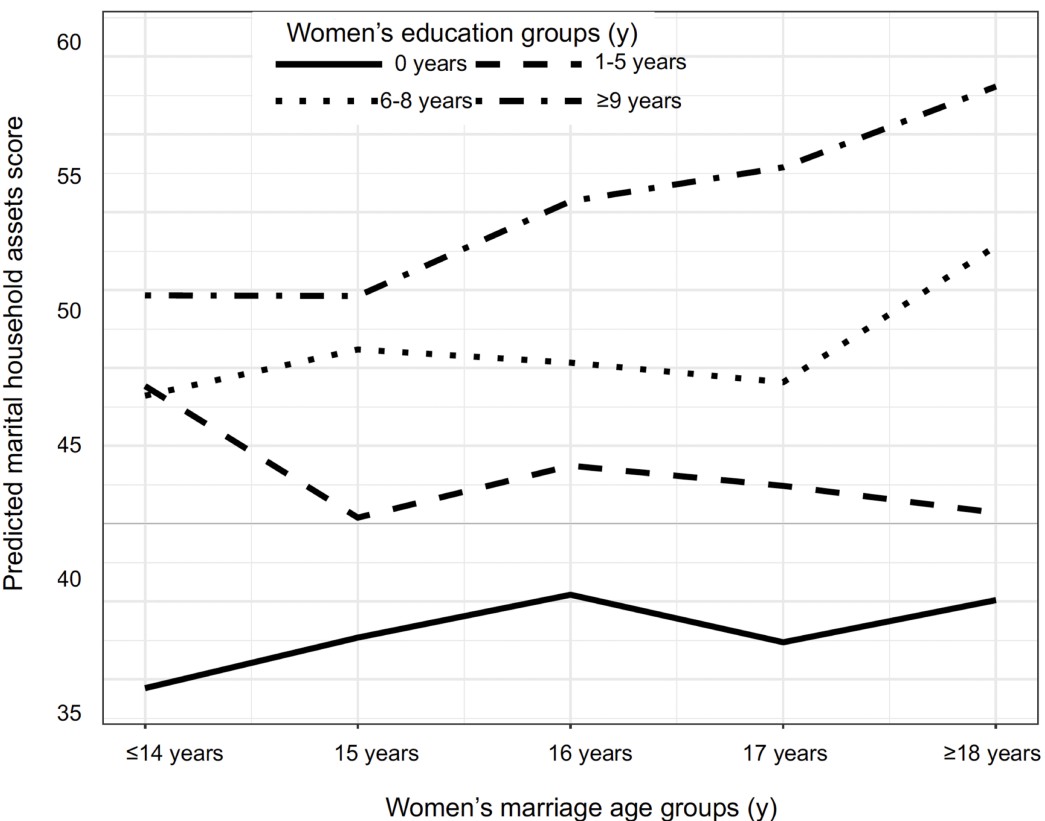

**Figure 4 Pay-offs in predicted marital household asset score stratified by women's marriage age and educational levels ($n$ = 3,102).** This scatterplot illustrates the standardized asset score ranging from 0–100. It parametrises Model 3 (in Table 2), and includes interactions between these two covariates. The model assumes a linear change in assets by marriage age group, and includes random effects by cluster (Table S2).

husband's education. Caste had a minimal effect and was statistically non-significant. Independent of education, each higher age-group for marriage was associated with 0.7% greater asset score, while compared to no education, completing primary, lower secondary or secondary/higher education was associated with 8.6%, 7.2% and 7.6% greater asset score respectively. The interaction effects, taking into account the above associations, indicate that each higher age-group for marriage was associated with 1.5% lower asset score for women with primary education, and with 0.3% and 1.3% higher asset score for those with lower secondary or secondary/higher education respectively. This model explained 29% of the variance in marital household asset score.

Table 3 Model 1 (hypothesis 3) focuses on only the uneducated women. It shows that relative to marrying ≤14 years, marrying at 15, 16, 17 and ≥18 years was associated with 2.7%, 4.9%, 3.2% and 6.8% greater marital asset score respectively. This model explained only 5% of the variance in marital household asset score. Model 2 includes husband's education and caste. Relative to marrying ≤14 years, marrying at 15, 16, 17 and ≥18 years was associated with 1.5%, 4.4%, 2.4% and 6.2% greater marital asset score respectively. Independently, each higher level of husband's education and caste affiliation was associated with a greater household asset score. Therefore, the independent effect of

**Table 3 Linear mixed-effects models of women's marriage age with marital household asset score for only uneducated women aged 12 to 34 years, surveyed within ≤1 year of marriage ($n$ = 1,223).**

| | Dep. Var. = Marital household asset score | |
| --- | --- | --- |
| | **Model 1**<br>**Women's marriage age** | **Model 2**<br>**Women's marriage age and marital household traits** |
| | $\beta$ *(standard errors)* | $\beta$ *(standard errors)* |
| Women's age (y) | −0.8 (0.5) | −0.7 (0.5) |
| Women's marriage age (y): ≤14 years | Reference | Reference |
| 15 years | 2.7 (2.0) | 1.5 (1.9) |
| 16 years | 4.9 (2.1)* | 4.4 (2.0)* |
| 17 years | 3.2 (2.4) | 2.4 (2.3) |
| ≥18 years | 6.8 (3.1)* | 6.2 (3.0)* |
| Husband's education (y): None | | Reference |
| Primary (1–5 years) | | 5.6 (1.5)*** |
| Lower-secondary (6–8 years) | | 6.4 (1.5)*** |
| Secondary or higher (≥9 years) | | 12.7 (1.7)*** |
| Caste: disadvantaged | | Reference |
| Middle | | 0.5 (1.2) |
| Advantaged | | 3.1 (1.5)* |
| Marginal $R$-squared | 0.01 | 0.10 |
| Conditional $R$-squared | 0.05 | 0.12 |

**Notes:**
$n$, number. Models include fixed and random effects estimates for geographic clusters.
* $p < 0.05$
*** $p < 0.001$.

women's own marriage age was partly mediated by marrying more educated husbands and /or those of more advantaged caste. This model explained 12% of the variance in marital asset score.

Overall, Table 2, Model 1 supports our first hypothesis, that women's later marriage age was associated with greater marital household asset scores. Model 2 supports our second hypothesis, that higher levels of education among women were associated with greater marital household asset scores. Models 3 and 4 support hypothesis 3, that the association of women's marriage age with the marital household asset score varied according to their level of education. This finding is due to the highly educated group showing a steeper increase in marital assets with marital age, compared to all other groups. Table 3 also supports hypothesis 3, by showing a very modest benefit in terms of asset wealth for delaying marriage for uneducated women beyond 16 years of age. Compared to marrying at 16 years, marrying at the earliest 'legal age' of ≥18 years was associated with a small additional payoff of ~1.8% greater asset score.

## Hypothesis 4

The assumption that each educational level is associated with a particularly common age at marriage is based on previous research suggesting a direct trade-off between marriage and schooling. For example, if a girl is in school, she is less likely to be married (*Sekine &*

(a)

| Marriage age | Uneducated (0y) | Primary education (1-5y) | Lower secondary education (6-8y) | Secondary or higher education (9+y) |
|---|---|---|---|---|
| ≤14 years | 144 | 38 | 89 | 50 |
| 15 years | 264 | 69 | 142 | 123 |
| 16 years | 346 | 112 | 182 | 257 |
| 17 years | 232 | 84 | 113 | 221 |
| ≥18 years | 237 | 57 | 54 | 288 |
| n | 1,223 | 360 | 580 | 939 |

(b)

| Marriage age | Uneducated (0y) | Primary education (1-5y) | Lower secondary education (6-8y) | Secondary or higher education (9+y) |
|---|---|---|---|---|
| ≤14 years | 11.8 | 10.6 | 15.3 | 5.3 |
| 15 years | 21.6 | 19.2 | 24.5 | 13.1 |
| 16 years | 28.3 | 31.1 | 31.4 | 27.4 |
| 17 years | 19.0 | 23.3 | 19.5 | 23.5 |
| ≥18 years | 19.4 | 15.8 | 9.3 | 30.7 |
| Total % | 100 | 100 | 100 | 100 |

**Figure 5 Heat maps illustrating (A) the overall numbers and (B) distributions of women getting married at a given age by education level.** Light blue shaded areas indicate low numbers/percentages, and dark blue shaded areas the highest numbers/percentages.

*Hodgkin, 2017*). Moreover, women who have attained more education tend to marry at an older age (*Guragain et al., 2017*; *Raj et al., 2014*; *Sekine & Hodgkin, 2017*; *Wodon et al., 2017*). Figure 5 presents heat maps illustrating (a) the overall numbers and (b) distributions of women getting married at a given age by education level. Light blue shaded areas indicate low numbers/percentages, and dark blue shaded areas the highest numbers/ percentages.

Heat map (A) shows that within the whole sample, women either completed lower-secondary or more education (≥6 years), or were uneducated, while few had completed only primary education. The common pattern was for women to be uneducated and to marry at 16 years, followed by secondary or higher educated women marrying ≥18 years. There are two potential explanations for the 50 women who had completed ≥9 years of education and were married at ≤14 years. First, they had enrolled in school at a young age, at ≤5 years, as found in other studies (*Marphatia et al., 2020b*). Second, women who were married at ≤14 years had continued their education after marriage, hence completing ≥9 years of schooling. Looking in more detail within education groups, heat map (B) shows that uneducated women tended to get married at 15–16 years, those with primary education at 16–17 years, those with lower-secondary education at 15–16 years, and those with secondary or higher education at 16 years and ≥18. This threshold effect of secondary

education, for marrying ≥18 years, has also been found in other studies (*Malhotra, Pande & Grown, 2003*; *Marphatia et al., 2020a*; *Pandey, 2017*; *Raj et al., 2014*).

These results provide partial support for hypothesis 4, namely that it was broadly most common to marry around the age when the maximum economic pay-off for the woman had been attained. For the highly educated group, marriage was commonest as expected at ≥18 years, when the economic benefits were greatest. For all other groups, the commonest age at marriage was around 15–16 years. We suggest that this pattern arises because, for all those women who were not highly educated, there would have been a negligible economic pay-off for delaying marriage past this age.

## DISCUSSION

Both public health researchers and feminist economists have long called for household data that better indicate how economic resources are unequally generated, controlled and allocated amongst family members, especially women (*Bradshaw, Chant & Linneker, 2017*; *Deere & Doss, 2006*; *Doss, Kieran & Kilic, 2019*). Our study extends this scholarship in the context of marriage patterns in a marginalized population in a low-income setting. Although Nepal has a legal minimum marriage age of 20 years (18 years with parental permission until recently) (*Government of Nepal, 2017*; *MOHP, New ERA & ICF International, 2017*), the Maithili-speaking Madhesi women of our study have the lowest median age at marriage (15 years) nationwide (*Marphatia et al., 2020a*), and marriage is well understood to involve financial considerations. Our analysis evaluated whether women would be more likely to enter more affluent marital households if they were married at later age. We found that later marriage (≥18 years) was associated with greater marital assets only for women with secondary school education. Conversely, for women with little or no education (defined as never receiving any formal education), there were very modest economic pay-offs in association with later marriage, and these women tended to be married at 15–16 years. Our findings may therefore help understand why underage marriage persists despite policy efforts and interventions to prevent it. We also highlight the need for clarity on how household asset wealth data are collected and used in research on underage marriage.

Many studies use marital wealth inappropriately, as a proxy for wealth in the natal household, to investigate whether poverty in that household predicts the risk of underage marriage. We have attempted to overcome this limitation by conducting two different analyses of the association of women's marriage age and wealth, using data on wealth that was collected in either the natal or marital household. In combination, these two analyses shed new light on the economic implications of variability in the age when women are married (Fig. 6).

Our earlier analysis, analysing assets in the natal household, showed that girl's lower education rather than household poverty was associated with being married early, and that other markers of socio-economic status (*e.g.*, agrarian land ownership, caste affiliation) were also not associated with early marriage (*Marphatia et al., 2021a*). Here, we further show that women married at later age (≥18 years) only married into households with more wealth if they also had secondary education. For all other levels of education, there was a

**(A) Findings from previous paper on same cohort as this study**

| Exposures | | Outcome |
|---|---|---|
| Girls' lower education | ⟶ | Girl's early marriage |
| Natal household poverty | ⟶X⟶ | |

**(B) Findings from this study**

| Exposures | | Outcome |
|---|---|---|
| Girls' lower education | ⟶ | Marital household poverty |
| Girls' early marriage | ⟶ | |

**Figure 6 Comparison of findings on wealth and marriage with previous article on cohort.** (A) illustrates findings from our previous article of women in the same cohort whose wealth was measured in their natal home. We investigated whether girl's lower education and natal household poverty (exposures) were associated with their early marriage (outcome). Results showed that girl's lower education was associated with marrying early, but not natal household poverty. (B) shows findings from this study, which investigated whether girl's lower education and marriage age (exposures) were associated with the wealth of the household that they marry into (outcome). We found that each later year of women's marriage was associated with 1.5% lower asset score for those with primary education, and with 0.3% and 1.3% higher asset score for those with lower secondary or secondary/higher education respectively.

negligible increase in marital household wealth in association with marriage age. Women were on average married at the earliest age when their marital assets would be the maximum expected for their level of education, which was at 15–16 years for all those without secondary education.

Any economic benefits gained through marriage may make meaningful differences in the daily lives of women, especially those who grew up in in poorer households, and might provide greater economic security to married women compared to remaining in their natal household (*Gram et al., 2018*). This may help understand why early marriage continues to be practiced in this population despite minimum marriage age and education legislation, with women typically being married at the earliest age that brings them the highest level of wealth that they could expect, according to their education. However, it is also likely that women maintain limited access to material resources, if gender disparities in asset ownership and wealth persist (*Bradshaw, Chant & Linneker, 2017*; *Deere & Doss, 2006*; *Doss, Kieran & Kilic, 2019*; *Swaminathan, Lahoti & Suchitra, 2012*).

Our findings have several implications for research and policy decision-makers. First, we need to revisit our academic understanding of the economic rationale for underage marriage. The economic value placed on women, and the 'bargaining' of their future by other household members, makes marriage a key feminist issue, which must be grappled with better at a theoretical level (*Agarwal, 1997*; *Seiz, 1995*). The feminist readership provides a crucial platform for giving voice to these marginalised women, and whose lack of education relates directly to patriarchal norms and customs. Feminist-rational choice theory (*Driscoll & Krook, 2012*), which has been applied in the context of marriage timing in the United States (*Cherry, 1998*), could be extended to populations in low-and middle-income countries. This approach could incorporate marital household wealth as an
element of the economic rationale underpinning underage marriage. This may better elucidate the trade-offs faced by families, which incorporate, among others, decisions over how much education, if any, to provide to daughters. However, any potential economic pay-off of early marriage needs to be squared with its adverse effects on maternal and child health and human capital outcomes (*Ganchimeg et al., 2014*; *Godha, Hotchkiss & Gage, 2013*; *Goli, Rammohan & Singh, 2015*; *Marphatia, Amable & Reid, 2017*), which may not be perceived by families when marrying their daughters.

Second, data that are more 'fit for purpose' will help us to monitor progress towards the SDGs more effectively, and also further understanding of the factors that drive early marriage. Third, women's education is a crucial factor associated with the type of family she ends up in through marriage, in terms of wealth. However, even among the highly educated women, the median age at marriage was 17 years (IQR 2.0) and 70% had married <18 years in our population. Thus, investment in the education of daughters may in part be driven by the contemporary marriage market (*Jeffrey & Jeffery, 1994*; *Maertens, 2011*), and in part by whether families are willing to, or able to afford sending their daughters to fee-paying private school or even public schools, which nevertheless incur costs related to learning materials and transport. Until and unless a collective shift in societal gendered norms takes place, 'where women get to' from their own education, and in terms of the assets of the marital home when they marry, will continue to be restricted to any economic security granted through marriage, rather than wider life opportunities (*Dixon-Mueller, 2008*).

Fourth, and from a policy perspective, school-based interventions to delay marriage will not reach the uneducated or out-of-school girls (*Marphatia et al., 2020a*). Therefore, equal effort is required to get girls into school in the first place, to keep them in school for longer, and to delay marriage. Out-of-school girls would also benefit from literacy and sexual and reproductive health information (*Mathur, Malhotra & Mehta, 2001*; *Verma, Sinha & Khanna, 2013*) provided through peer or community groups (*Marphatia & Moussié, 2013*; *Prost et al., 2013*).

Our study includes some limitations: while we were able to investigate the associations between marital asset score and women's marriage age and their education, we could not establish causality with cross-sectional data. From our data, we could not deduce whether women were married because they had dropped out of school, or they stopped studying because they were married. However, previous studies from rural India have found that, although both of these situations are possible, most girls tend to drop out of school first, and are then married (*Marphatia et al., 2022b*). As in a previous study (*Marphatia et al., 2022b*), Fig. 5 in our analysis suggests that a minority of women ($n = 50$) may have continued their education after marriage, or they may have started school at a very young age, as shown elsewhere (*Marphatia et al., 2020b*).

Another limitation is that our asset score was measured in the marital household after, and not at, marriage. To address this, we excluded items which may have been purchased after marriage from our total asset score and restricted the sample to women surveyed within 1 year of marriage. Similarly, wealth data in our cohort were collected in either the natal or marital households, but not both. We therefore could not include natal household

wealth in regression models and could not deduce whether women were in fact marrying into households that were wealthier than their natal homes. Our goal in this article, in the absence of knowing the wealth of the household that women came from, was simply to assess whether women ended up in a household with greater wealth if they married later. We also did not collect data on dowry, which may increase the average wealth in marital households.

Another bias in our sample was the recruitment of only pregnant women. A previous analysis comparing the women in our study with a random sample included in the DHS of all non-pregnant women from the same region in Nepal found that our sample was younger and lower educated, and had fewer children; a second comparison between our sample and currently married and pregnant women from the same region found our sample was younger and less educated, but there were no differences in their reproductive history (*Marphatia et al., 2020a*). Our sample may not be representative of all older women. Finally, caution is needed when extrapolating our results beyond other Terai or Maithili-speaking Madhesi populations in North India, given the early marrying, lower educated and disempowered profile of our sample. We are nevertheless confident that our appropriate use of household wealth data yields robust associations of women's marriage age with the marital asset score, and these will be widely applicable. Our results may help understand the economic rationale underpinning the timing of marriage, and why early marriage remains common despite efforts to delay it.

## CONCLUSIONS

Surveys incorrectly use wealth markers in the marital household as a proxy for natal wealth to investigate whether poverty precipitates women's underage marriage. Our previous analysis of the same cohort investigated the association of natal household wealth with women's age at marriage and found that lower education, rather than poverty, was associated with early marriage (*Marphatia et al., 2021c*). In this article, we used marital wealth data more appropriately, to provide insight into a new question: on the potential economic consequences, in terms of marital wealth, of women marrying at different ages. In rural lowland Nepal, highly educated women were in households with substantially greater marital assets if they were married later, whereas for women with little or no education, later marriage was associated with very modest pay-offs. The lack of substantive economic pay-off for delaying marriage among the majority of the women may help explain why the practice remains widespread despite minimum marriage age legislation. A societal shift in gendered norms is required to ensure that women can apply their education to access wider life opportunities other than the economic security granted through marriage.

## ACKNOWLEDGEMENTS

We would like to thank the women and their families for participating in the LBWSAT study, and for permitting us to measure their newborns. We are grateful to the Public Health Offices of Dhanusha and Mahottari Districts in Nepal, who supported implementation of the trial. Data collection was facilitated by the following staff of Mother

and Infant Research Activities (MIRA): Bhim Prasad Shrestha, Aman Sen, Sonali Jha, Anjana Rai, Raghbendra Sah, Puskar Paudel, Bishnu Bhandari, and Rishi Neupane. We would also like to thank the 66 data collectors, their 16 supervisors, and 720 volunteer enumerators. We also acknowledge the University College London Institute for Global Health team, including Sarah Style, Helen Harris-Fry, B. James Beard, Andrew Copas, Joanna Morrison, Lu Gram, Jayne Harthan, Jolene Skordis-Worrall, Anni-Maria Pulkki-Brannstrom, David Osrin and Anthony Costello.

### Funding

This research was supported by the Leverhulme Trust (Grant Number: RPG-2017-264) and National Institute for Health Research (NIHR) Great Ormond Street Hospital Biomedical Research Centre. Funding for the LBWSAT was provided by the Department for International Development (DFID) South Asian Research Hub (Grant Number: PO 5675). The funders had no role in study design, data collection and analysis, decision to publish, or preparation of the manuscript.

### Grant Disclosures

The following grant information was disclosed by the authors:
Leverhulme Trust: RPG-2017-264.
National Institute for Health Research (NIHR) Great Ormond Street Hospital Biomedical Research Centre.
Department for International Development (DFID) South Asian Research Hub: PO 5675.

### Competing Interests

The authors declare that they have no competing interests.

### Author Contributions

- Akanksha A. Marphatia conceived and designed the experiments, analyzed the data, prepared figures and/or tables, authored or reviewed drafts of the article, and approved the final draft.
- Naomi M. Saville conceived and designed the experiments, performed the experiments, analyzed the data, authored or reviewed drafts of the article, and approved the final draft.
- Dharma S. Manandhar conceived and designed the experiments, performed the experiments, authored or reviewed drafts of the article, and approved the final draft.
- Mario Cortina-Borja conceived and designed the experiments, analyzed the data, prepared figures and/or tables, authored or reviewed drafts of the article, and approved the final draft.
- Jonathan C. K. Wells conceived and designed the experiments, analyzed the data, prepared figures and/or tables, authored or reviewed drafts of the article, and approved the final draft.

## Human Ethics

The following information was supplied relating to ethical approvals (*i.e.*, approving body and any reference numbers):

The Nepal Health Research Council (108/2012) and University College London (UCL) Research Ethics Committee (4198/001) granted ethical approval to conduct the LBWSAT. Additional ethical approval was obtained from the Research Ethics Committee at UCL (0326/015), the University of Cambridge (1016), and the Nepal Health Research Council (292/2018) for secondary analyses of LBWSAT data.

## Data Availability

Raw data are available in the Supplemental Files.

## Supplemental Information

Supplemental information for this article can be found online at http://dx.doi.org/10.7717/peerj.17671#supplemental-information.

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
