# Peer review of "Where have I got to? Associations of age at marriage with marital household assets in educated and uneducated women in lowland Nepal"

_PeerJ, doi:10.7717/peerj.17671_

## Round 0.1 · original submission · Major Revisions

Please respond carefully to the reviewers' comments, particularly on the statistical analysis. Please also ensure that the manuscript is clear and easy to understand by receiving appropriate scientific editing.

**Language Note:** The Academic Editor has identified that the English language must be improved. PeerJ can provide language editing services - please contact us at [email protected] for pricing (be sure to provide your manuscript number and title). Alternatively, you should make your own arrangements to improve the language quality and provide details in your response letter. – PeerJ Staff

Reviewer 1 ·

Basic reporting

Thank you for undertaking this valuable project. I have read the paper with great interest. I have found that the paper presents some important finding on associations of age at marriage with marital household assets in educated and uneducated women from lowland Nepal. I believe the paper can be accepted in PeerJ after some references check.

Experimental design

The research design used to explain the findings is broadly appropriate. The model explains the associations of age at marriage with marital household assets in educated and uneducated women properly in my opinion.

Validity of the findings

The authors have presented the findings properly and the results appear valid statistically. The linear mixed-effects regression models used to calculate the marital asset score is appropriate mostly.

Additional comments

None.

·

Basic reporting

Good

Experimental design

1. This study uses linear mixed-effects regression models to answer the research questions. The author needs to explain why the ordinary least square (OLS) regression model is not adequate to answer the research questions such that a more complex model is needed to answer the research questions.

2. Like any statistical model, linear mixed-effects models have some assumptions that need to be checked before drawing conclusions from the results. The authors need to explain whether these assumptions have been satisfied in their study.

Validity of the findings

3. In general, the conditional R2 is higher than the marginal R2, because it considers both the fixed and random effects, while the marginal R2 considers only the fixed effects. In the manuscript, the authors report only the conditional R2 statistic (i.e., the proportion of variance explained by both fixed and random effects) for their mixed-effects models with the conditional R2 values of their models ranging from 0.05 to 0.29.
(a) Does the analysis presented in the manuscript based on models with relatively small conditional R2 values reasonable? Please provide explanation regarding this matter.
(b) I recommend that both marginal and conditional R2 be reported given that they convey unique and distinctive information.
(c) Since the R2 has some limitations, especially for mixed-effects models, which include both fixed and random effects, I recommend that some alternatives of model evaluations in mixed-effects models besides R2 be presented.

4. In Table 4 Heat Map 1, in the first row and the fifth column, there are 50 women who have completed 9 years of education or more and were married at 14 years of age or less. There are two possible interpretations that come to my mind for this assuming the number presented is accurate. First, their first enrollment to attend school was at five years old or less. Second, women who were married at the age of 14 or less continued their education that they can achieve 9 years of schooling or more. Do these make sense? The authors need to clarify this matter.

Additional comments

5. In Table 2 for Model 3 (column 4) and Model 4 (column 5), the regression coefficients for women's age is 0.0. I suggest that the decimal scale be set more digits (up to 3 or 4 decimal scale) that readers could see the value of the regression coefficient for this variable.

6. I wonder, could it be known from this study whether the women married because they dropped out of school or they dropped out because they wanted to marry?

Thank you and Good luck.

Reviewer 3 ·

Basic reporting

The authors have done a good job using clear language, have cited relevant studies, and have reported results relevant to their hypotheses.

I have one lingering question that the authors may wish to address when describing the Nepali context. Is education in this area mostly public, private, or a mix of these? If, say, the only secondary school options are private, these could be quite expensive, indicating high natal household wealth for those women. Outlining how education is structured in this part of Nepal may help you justify why education is a better natal household wealth proxy than marital household wealth.

Experimental design

This research is very interesting and important. I thought the authors did a very clear job outlining their hypotheses, methods, and results. However, I think there are some changes to the framing of the paper that could make it stronger.

Much of the introduction focuses on how marital household wealth has inappropriately been used as a proxy for natal household wealth. I think this information is relevant to mention, but overall tangential to your hypotheses. I think lines 159-177 should be moved forward in the introduction because your main goal is to understand why individuals may enter into early marriages. The discussion of how natal household wealth is associated with marriage timing should be reduced because these analyses are not using natal household wealth, nor do they speak to those relationships between natal wealth and marriage timing. I think it is appropriate to briefly state that previous studies have incorrectly used marital wealth as a proxy for natal wealth, but that your study will instead use education as a proxy and examine marital household wealth as an outcome. I think refocusing the intro and lit review in this way will make the hypotheses clearer and the result more impactful.

The methods were explained well.

Validity of the findings

The authors have done a good job explaining and presenting their results. The figures and tables are readable and present relevant information. The data are provided.

I have a couple questions about the data that the authors may want to address.
1. I find the sample to be a bit odd to begin with because these are all women who are pregnant within the first year of marriage. I'm wondering if this biases the types of individuals in their data set in any meaningful way. For example, highly educated women may wish to delay pregnancy, especially if they marry into a poorer household but expect to increase their wealth by working for a few years prior to having children. The authors could address this a few different ways: They could compare the demographics of the women who did and did not get pregnant during the monitoring period (if the data is available), they could comment about the typical time from marriage to first birth/pregnancy in the area, and/or, if none of the other options are possible, they should simply acknowledge this as an unknown source of bias in their sample.

Additional comments

Overall, I think the article is very sound. I recommend some revisions to the framing of the paper, especially the introduction/lit review, and just had a couple points for the authors to clarify about the sample/Nepali context they are examining. These suggestions do not require major revisions of the hypotheses, analyses, or conclusions, so I think the paper is nearly there! Good work!

Reviewer 4 ·

Basic reporting

.

Experimental design

.

Validity of the findings

.

Additional comments

Overview:

Women’s early marriage is a global health issue as it leads to adverse
maternal and child health outcomes. It also impact women’s education and employment
outcomes. Most studies use poverty in women’s natal household to predict the risk of
women’s early marriage. The paper points out the absence of using women’s marital
household wealth as a predictor to form an interesting research gap. Using data from
the cluster-randomised Low Birth Weight South Asia Trial (LBWSAT) and employing a
linear mixed-effects model with random effects, the paper broadly seeks to identify the
characteristics of women who are at the highest risk of getting married below the legal
age in Nepal. To do so, the authors pose two research questions. First, what is the
association between marital household asset and age at marriage for women? Second,
does this relationship vary by women’s education level? The authors find that marriage
amongst women aged 18 years or above is associated with greater marital assets for
secondary-educated women, while there is only a modest increase in assets for
uneducated women below 16 years of age. The identified research gap and findings are
interesting and of great policy importance in addressing women’s early marriage and
related gender disparities.

Major Comments

1) Contribution and Literature review – The contribution of this paper is unclear
since the authors have multiple papers probing similar research questions. They
even have a paper in this journal where they study the association between natal
household wealth and women’s early marriage. Interestingly, they the use the
same setting and dataset in these studies. It would be useful if the authors
discuss their previous findings and highlight how this paper builds on their
previous analyses. Preferably, my recommendation would be to write one paper
using age at marriage as the dependent variable and conduct bivariate and
multivariate regression analysis to study all the possible covariates of marriage
age at once, such as natal household wealth, marital household wealth, husband’s
education, distance between natal and marital home, wealth gap between marital
and natal home, husband’s employment and earning etc. Else, the contribution
should be made clearer to add to the credibility of the paper.

2) Marital household wealth measure – There are three concerns about the marital
household asset score. First, the authors use principal component analysis (PCA)
to construct the marital household asset score, which is a well-accepted measure
for index creation. However, the authors use the first principal component as
their index measure that explains only 34.5% of the variation in marital
household wealth. Therefore, as a robustness check, I recommend that the
authors create an individual asset score and then create a binary outcome using
the mean or median assets owned. Alternatively, they can justify the accepted
range of variation in PCA literature so that their household wealth index seems
more acceptable. Second, what does the measure signify as a standalone variable
of interest? If it signifies a better match for the women, we cannot be sure that
the women are in fact marrying into wealthier families unless the authors
control for natal wealth. So, the authors should include natal wealth in the
regression. Third, the authors may be capturing an increase in marital household
wealth due to reasons other than the studied and control variables. This increase
could be due to dowry wealth received at the time of marriage as there is a wide
practice of dowry in South Asian countries, including Nepal. Perhaps, the authors
could state that dowry wealth may increase the average wealth across all marital
households, or potentially impact their results and add it as a limitation.

3) Additional robustness checks: The authors should cluster standard errors at the
village/cluster level because the responses of women from a cluster/village may
be correlated. Clustering of standard errors will likely increase the standard
errors and it is crucial to check if the results persist after doing so.

4) Discussion: The authors rightly identify policy implications of the association
between education and delayed marriage. Additionally, it is possible that despite
marrying into a wealthier household, gender disparities in asset ownership and
wealth continue to exist (see Swaminathan, Lahoti and Suchitra, 2012). The
authors can mention this topic as needing further research.

Minor comments

1) File attachments – It states that there are 4 figure and 6 table files, but there are
only 3 figures and 4 tables in the paper. The authors are requested to kindly
check if there is any discrepancy.

2) Paragraph formation – The authors should ensure that each paragraph has at
least three sentences and each para should have a first/topic sentence explaining
the information in the paragraph.

3) Title of the paper – The authors can do away with “Where have I got to?” from
the title to keep it short and direct.

4) Reporting language – Although the authors acknowledge the lack of causal
evidence as a study’s limitation, it is recommended that the authors carefully
consider the language used while reporting their findings in other sections.
discover an association. For instance, the abstract conclusion states “On average,
marrying 18 years yields greater marital assets for secondary-educated women”,
which suggests causality instead of correlation. Similarly, in line 245, Hypothesis
4 states “that for any given level of education, women are most commonly
married at an age which would maximize marital household assets”, implying
that the early marriage decision is made because of economic advantage.

5) Grammatical errors – In line 212, ‘is’ has been wrongly formatted in the sentence
case. In line 326, the word ‘they’ has been incorrectly added.

Conclusion

The paper provides a reasonable reconsideration of the association between household
wealth, women’s early marriage and women’s education in Nepal. It highlights an
important aspect in existing literature, that is, marital household wealth as an under
researched topic in women’s early marriage. The topic is interesting and important.
However, the paper does not make a clear contribution to literature and misses out on
additional covariates that could potentially affect the results.
In my view, revisiting the literature review and empirical strategy can strengthen the
paper’s credibility.

---

## Round 0.2 · Minor Revisions

The authors were able to respond to many comments, but several minor revisions are needed. Please address them in this final revision round

Reviewer 1 ·

Basic reporting

The paper looks much improved now including the methods section where major concerns were addressed.

Experimental design

The design is concurrent with the study objective.

Validity of the findings

Results look valid, and the correct yardsticks were used.

·

Basic reporting

Good

Experimental design

Good

Validity of the findings

Good

Additional comments

I appreciate your efforts to respond to all questions and suggestions.
The revised manuscript is much better than the previous version.
I look forward to the publication of this manuscript.

Reviewer 3 ·

Basic reporting

Overall, the paper has all the elements needed. However, there are several places where the writing is unclear or where the ideas presented seem disorganized. Some paragraphs are also very short and could be combined with each other. I would encourage the authors to do a final reading for clarity, conciseness, and organization. I have provided some line edits below to facilitate this, but the entire manuscript should be carefully read one more time before publication.


Line edits:
Intro, Line 120: Change “Moreover” to “Despite this” to better connect this sentence to the point of the paragraph.

Literature review line 126- 128: Right now you point out two examples where authors incorrectly used wealth measures and find no/inconsistent associations, but there is no information on what outcome these studies were testing.

Hypotheses: The hypotheses don’t need to start with the word “that”.

Line 339: “Scatterplots show…” Are these provided in the paper or as supplemental material? If so, they need to be referenced here. If not, clarify that you just inspected scatterplots separately.

Lines 468-471: This section is confusing as written and contains information that is redundant with the preceding paragraph. A much simpler phrasing would be something like: “Figure 3 shows the predicted values derived from Model 3 for marital household asset score by women’s marriage age and education.” That sentence can then just be the first sentence of the next paragraph.

Line 502: Move “Model 1 (hypothesis 3)” to the beginning of the next sentence. It is confusing as written.

527-531: Tell the readers exactly what pattern in the heat maps would support this hypothesis. Currently, you don’t connect the heat map to the marital asset score results until line 555, and before I got there I wasn’t sure what I was supposed to take away from Table 4.

Table 4: Are both a and b necessary? The patterns seems very similar, so the authors might consider just presenting one heat map. I would suggest just keeping b, but this is ultimately up to the authors.

534-535: The text and table 4 caption say the numbers are highlighted red and green, but they are now varying shades of blue. Please correct.

655-660: Some of the information in this paragraph could be moved to the methods and omitted from the discussion. I do not see the relationship between the last sentence and the methodological issues with collecting accurate age data. This sentence should be deleted or moved to a more suitable location in the paper.

662-667: I am not sure this sentence addresses the issue with only recruiting pregnant women, possibly because the writing is confusing. The goal is to compare the sample of pregnant women in this study to a sample of non-pregnant married women in the same area to see if there are substantial differences. Based on the reviewer comments, you may have done that, but it is still not clear to me. The comparison you offer here with DHS data does not clarify whether you are comparing married pregnant women or married women who are not pregnant. I’m also not sure what getting sterilized later in life has to do with this study, and I think this could be removed.

Figure 4: I do not think this figure is necessary. The written discussion of these results in the text is sufficient.

Experimental design

Research questions are clear. Analyses are sufficient and are described clearly.

Validity of the findings

The conclusions are mostly clear. The analyses and results support the conclusions drawn.

Additional comments

The hypotheses, methods, and results are sufficient for publication. In my opinion, the writing just needs some minor improvements for clarity and conciseness. See line edits above.

Reviewer 4 ·

Basic reporting

I have read through the manuscript and the revisions made by the author. They have included most of my requested suggestions and made appropriate changes to the manuscript.

Experimental design

The revisions in this section are appropriate.

Validity of the findings

The revisions in this section are appropriate.

---

## Round 0.3 · accepted · Accept

Dear authors, after these last revisions, your work is now acceptable for publication. Please, proceed in the proofreading stage to fix small issues such as referencing and consistency in referencing (eg. "https://doi-org.libproxy.ucl.ac.uk/10.1080/19452829.2016.1251403. - will block us with a UCL library wall; using http or not in the DOIs , etc...)